# Revisiting the Large-Conductance Calcium-Activated Potassium (BKCa) Channels in the Pulmonary Circulation

**DOI:** 10.3390/biom11111629

**Published:** 2021-11-03

**Authors:** Divya Guntur, Horst Olschewski, Péter Enyedi, Réka Csáki, Andrea Olschewski, Chandran Nagaraj

**Affiliations:** 1Experimental Anaesthesiology, Department of Anaesthesiology and Intensive Care Medicine, Medical University of Graz, Auenbruggerplatz 5, 8036 Graz, Austria; divya.guntur@medunigraz.at; 2Department of Internal Medicine, Division of Pulmonology, Medical University of Graz, Auenbruggerplatz 15, 8036 Graz, Austria; horst.olschewski@medunigraz.at; 3Ludwig Boltzmann Institute for Lung Vascular Research, Neue Stiftingtalstraße 6, 8010 Graz, Austria; Nagaraj.Chandran@lvr.lbg.ac.at; 4Department of Physiology, Semmelweis University, Tűzoltó utca 37-47, 1094 Budapest, Hungary; enyedi.peter@med.semmelweis-univ.hu (P.E.); csakireka2@gmail.com (R.C.)

**Keywords:** large conductance calcium-activated potassium (BKCa) channels, KCNMA1, KCNMB1, KCNMB2, LRRC26, pulmonary circulation, hypoxia, fetal to neonatal transition, pulmonary hypertension

## Abstract

Potassium ion concentrations, controlled by ion pumps and potassium channels, predominantly govern a cell′s membrane potential and the tone in the vessels. Calcium-activated potassium channels respond to two different stimuli-changes in voltage and/or changes in intracellular free calcium. Large conductance calcium-activated potassium (BKCa) channels assemble from pore forming and various modulatory and auxiliary subunits. They are of vital significance due to their very high unitary conductance and hence their ability to rapidly cause extreme changes in the membrane potential. The pathophysiology of lung diseases in general and pulmonary hypertension, in particular, show the implication of either decreased expression and partial inactivation of BKCa channel and its subunits or mutations in the genes encoding different subunits of the channel. Signaling molecules, circulating humoral molecules, vasorelaxant agents, etc., have an influence on the open probability of the channel in pulmonary arterial vascular cells. BKCa channel is a possible therapeutic target, aimed to cause vasodilation in constricted or chronically stiffened vessels, as shown in various animal models. This review is a comprehensive collation of studies on BKCa channels in the pulmonary circulation under hypoxia (hypoxic pulmonary vasoconstriction; HPV), lung pathology, and fetal to neonatal transition, emphasising pharmacological interventions as viable therapeutic options.

## 1. Introduction

Several mechanisms regulate the pulmonary arterial vascular tone. On the one hand, the potassium (K^+^) channels that control membrane potential in endothelial and smooth muscle cells directly influence pulmonary arterial tone. On the other hand, potent relaxing and contracting agents produced and released by endothelial and circulating cells, control the tone of the adjacent smooth muscle cells. All these local or circulating stimuli frequently act on different K^+^ channels of endothelial and smooth muscle cells, underpinning the crucial role of these unique membrane structures in the pulmonary circulation [1,2]. Furthermore, they are also targets for several redox-active signalling factors that are circulating and locally produced [3]. Mutation of genes encoding K^+^ channels and the subsequent loss-of-function of these channels or their chronic inhibition, either directly or indirectly by targeting their regulatory mechanisms, has the potential to result in chronic pulmonary vasoconstriction and vascular remodelling with thickening and stiffening of the vessel wall [4,5]. This review focuses on the contribution of the large conductance calcium (Ca^2+^)-activated K^+^ channel (BKCa). To better understand the role of this channel, it is worth looking at the blood flow in the lung first.

## 2. The Pulmonary Circulation, a System with Many Unique Features

There are some significant differences between the pulmonary arteries and the systemic arteries. Compared to the systemic circulation, the pulmonary circulation is a low-pressure–high-flow system. The mean pulmonary arterial pressure (Pap) for humans at rest is 14.0 ± 3.3 mmHg, approximately 15% of the systemic circulation [6]. Pap is independent of sex, posture, and geographical origin, except highlanders, and only slightly dependent on age with an upper limit of normal below 21 mmHg [6]. A significant anatomical difference between the pulmonary and systemic circulation is the low pulmonary vascular resistance (PVR), about 10 times lower than the systemic vascular resistance, and that under physiologic conditions the autonomous nervous system has limited control over resting and exercise PVR [7] The normal pulmonary vasculature has no fully muscularised arterioles. Instead, the large, muscular arteries branch into small, partially muscular vessels with a meagre perfusion resistance and directly feed the pulmonary capillaries. The capillaries build a network within the alveolar wall where the gas exchange between the alveolar gas and blood occurs.

The pulmonary arteries respond to a drop in the alveolar oxygen tension with a rapid and reversible contraction, which is called “hypoxic pulmonary vasoconstriction” (HPV) [8,9,10]. The intensity of this reaction is proportional to the degree of hypoxia, and it occurs independently of neural or humoral factors [11]. HPV is a conserved response and starts within seconds of the onset of alveolar hypoxia as soon as the alveolar oxygen tension falls by more than ~10% [12,13,14]. HPV allows optimizing the ratio between ventilation and perfusion in the lung and prevents hypoxaemia due to ventilation/perfusion mismatch during the postnatal life [15]. However, if a critical mass of the lung becomes hypoxic, as seen in many lung diseases and in high-altitude exposure, the subsequent pulmonary vasoconstriction and chronic pulmonary arterial remodelling contributes to pulmonary hypertension, right heart failure, and finally premature death. According to current knowledge, oxygen-sensitive ion channels such as potassium (K^+^) channels of the pulmonary arterial smooth muscle cells are widely accepted to represent the executive for the regulation of the vascular tone in response to hypoxia [16]. Furthermore, HPV is modulated by acidosis and alkalosis [17]. Since in human pulmonary arterial smooth muscle cells the pH-sensitive two-pore domain channel TASK-1 has a major impact on the resting membrane potential, it is conceivable that these K^+^ channels play an essential role in the pH-driven modulation of HPV [18].

At rest, the pulmonary arteries are nearly maximally vasodilated and there is no evidence for significant contribution from the autonomic nervous system. Increased sympathetic activity leads to the release of catecholamines (epinephrine, norepinephrine, and phenylephrine) and NPY, resulting in vasoconstriction that may contribute to an increase in pulmonary vascular resistance and remodelling. In the last century, considerable advances have been made in the understanding of the regulation of cardiac K^+^ channels by catecholamines [19,20], whether the effects of these humoral factors in the pulmonary circulation are directly linked to K^+^ channel activity has not been elucidated yet. Local or circulating humoral factors influencing pulmonary arterial pressure have been extensively studied during the last decades. Endothelin-1 (ET-1), serotonin (5-HT), angiotensin II (ANG II), and arachidonic acid metabolites (prostaglandins) are the most extensively investigated vasoconstrictors. They are produced either by the endothelium of the pulmonary vessels or secreted from circulating cells. NO, atrial natriuretic peptide (ANP), vasoactive intestinal peptide (VIP), prostacyclin, alkalosis, and adenosine are known to be potent pulmonary vasodilators but this becomes only evident if the pulmonary pressure is elevated. For some of these factors, BKCa is likely to be the main target in regulating the pulmonary tone.

Finally, there is another remarkable complex physiologic process where K^+^ channels seem to have an essential role in the pulmonary circulation—the transition from the fetal to the neonatal circulation. The intrauterine to extrauterine circulation adaptation requires concerted but opposite action of the pulmonary arteries and the Ductus arteriosus Botalli. The pulmonary vascular resistance decreases immediately, whereas the fetal extracardiac shunt pathways, including the Ductus arteriosus Botalli, constrict and close. In both fetal pulmonary and Ductus arteriosus Botalli smooth muscle cells, BKCa and several voltage-sensitive (Kv) potassium channels have been identified over the past decades. Employing functional investigations of these channels by the patch-clamp technique, their activity has been frequently shown to depend on changes in oxygen tension or on the redox state [3]. In human Ductus arteriosus Botalli Kv, channels have been reported to modulate vessel tone. The switch to normoxia results in a reduction of the whole-cell Kv current, subsequent depolarisation of the cells, and vasoconstriction [21,22]. In contrast, in fetal lambs, activation of BKCa seems to be the driving force for oxygen and nitric oxide-induced perinatal pulmonary vasodilation [23,24]. Specifically, oxygen causes a localized release of Ca^2+^ from a ryanodine-sensitive intracellular Ca^2+^ store in neonatal PASMCs, which leads to the activation of the BKCa channel, generation of spontaneous transient outward potassium currents, membrane hyperpolarization, and vasodilation [24].

## 3. Structure and Gating of the Large-Conductance Calcium (Ca^2+^)-Activated Potassium (BKCa) Channels

BKCa channels belong to the heterogeneous family of the Ca^2+^-activated K^+^ channels. Like most cells in the body, K^+^ channels are the dominant ion channels in the plasma membrane of pulmonary arteries′ endothelial and smooth muscle cells, substantially contributing to the resting membrane potential. Due to the high input resistance of the plasma membrane of both cells (in the order of 10 GΩ), small changes in steady-state membrane current are sufficient to produce significant changes in membrane potential [25,26]. Based on the K^+^ electrochemical gradient, activation of the channels results in K^+^ efflux from cells and membrane hyperpolarisation. Conversely, inhibition of open K^+^ channels leads to membrane depolarisation.

Extended investigations from the last decades show that cells also express intracellular BKCa. Mitochondrial BKCa were the first to be identified [27] and now they are the best established amongst the intracellular BKCa channels. In isolated mitochondria from brain and heart, mitochondrial BKCa is known to reduce reactive-oxygen species production [28]. The activation of the channels with NS11021 improved cardiac mitochondrial function by enhancing the potassium uptake without a significant change in mitochondrial membrane potential, resulting in cardioprotection in an animal model [29]. BKCa currents were recorded in the outer membrane of the nucleus in rat pancreatic acinar cells [30]. Studies in endothelial cells indicate that nuclear BKCa is coupled to the activity of perinuclear prostaglandin receptors (EP3), regulating the nuclear membrane potential and expression of endothelial nitric-oxide synthase [31]. BKCa channels are synthesized in the endoplasmic reticulum [32]. Whether the BKCa channels are active in the endoplasmic reticulum is not yet established. Finally, since the Golgi apparatus is essential for post-translational modifications of channel proteins, it could be assumed that functional ion channels are localized here. In fact, several chloride channels have been shown to be active in the Golgi apparatus, but so far no functional K^+^ channel has been identified. The important question is what the function of intracellular BKCa might be. In analogy to the plasma membrane BKCa, it can be assumed that intracellular BKCa channels are also working as Ca^2+^ sensors in intracellular organelles. The flow of potassium ions via such channels could be important for maintenance of the ionic homeostasis for subcellular functions. Finally, since these channels interact directly or indirectly with other proteins, intracellular BKCa could be playing a role as a signalling molecule [33].

BKCa is coded by a single gene (KCNMA1 in humans or Slo1 in mouse) that undergoes extensive pre-mRNA splicing. The channel activity is regulated by numerous mechanisms. On the transcriptional level, alternative splicing influences Ca^2+^ sensitivity, channel kinetics, localization, and hormone sensitivity [34]. Co-assembly with different subunits changes the channel′s Ca^2+^ and voltage-sensitivity, as well as pharmacological properties [35,36,37,38,39,40,41]. On the post-translational level, channel activity is regulated by a number of endogenous mediators such as arachidonic acid, NO, pH and phosphorylation of the channel by protein kinase A, C, G, and CaMKII (Figure 1). Finally, co-localization with other macro-molecular signalling complexes including Ca^2+^-selective channels or close proximity to the IP3-receptors and ryanodine receptors adds a regulatory level.

The channel is formed by tetramers consisting of four monomers known as the α-subunits. Each of the four α-subunits is made up of seven transmembrane segments (S0–S6), of which S1–S6 show sequence similarity with another large group of K^+^ channels—the voltage-gated K^+^ channels. These transmembrane segments are responsible for the voltage sensitivity of the channel. S1–S4 domains sense the voltage changes while the S5 and S6 domains and the pore forming loops of the four monomers assemble the channel pore—the conduction pathway. The S0 segment is unique to the BKCa channel and builds the extracellular N-terminus responsible for binding to modulatory β-subunits. Another salient feature of this channel is the presence of the S7–S10 segments in the intracellular region—the C-terminus of the protein. S7–S10 is twice as long as the transmembrane part in terms of primary amino acid sequence and the last two segments S9 and S10 have a highly conserved inter-species sequence. This C-terminus creates a tetrameric gating ring containing two high-affinity Ca^2+^ binding sites. BKCa can co-assemble with modulatory β- and γ-subunits to modify the channel functional properties and pharmacology. Up to four β-subunits (β1–β4) have been identified during the last decades. The β1-subunit seems to be crucial for controlling the smooth muscle tone, as β1 can increase the apparent Ca^2+^ sensitivity of the channel. The discovery of the γ subunit (γ1–γ4) made BKCa channels important to cells in which the internal Ca^2+^ is not subject to significant changes.

The gating of the BKCa is complex, it can be activated by membrane depolarisation or intracellular calcium [Ca^2+^], separately or by both factors synergistically [42]. Recent allosteric modulations assume four modular voltage sensors and four Ca^2+^ binding sites [43]. Activation of these sensors can independently lead to channel opening and communication between the sensors and the pore and between sensors. Ca^2+^ binding leads to a leftward shift of the steady-state open probability of BKCa channels. The affinity to Ca^2+^ has been determined to be between 1 and 10 µM, but regulated additionally by modulatory proteins and metabolic conditions. BKCa is often co-localised with Ca^2+^ sources. They are voltage-gated Ca^2+^ channels (VGCC) or transient receptor potential vanilloid (TRPV) channels. In addition, functional coupling between BKCa in the plasma membrane and ryanodine receptors in the sarcoplasmic reticulum is critical for well-regulated smooth muscle functions, including the regulation of vascular tone and blood pressure [44]. The co-localization with such Ca^2+^ sources is important because raising the channel open probability to reasonable values (*p*_o_ ≥ 0.5) at membrane voltages in the range −50 to 0 mV requires a range of Ca^2+^ concentrations up to 10 μM or even more. Moreover, cytosolic Ca^2+^ per se also has the ability to open the channels. There is a lack of information on the regulation of BKCa by ions other than Ca^2+^ although studies suggest that Mg^2+^ increases the Ca^2+^ binding to the protein [45]. Nevertheless, Mg^2+^ is incapable of substituting Ca^2+^ for the functional coupling of the α and the β-subunits and thereby the channel activation [46]. The channel is exclusively voltage-gated at intracellular free Ca^2+^ concentrations below 100 nM, attributed to the dissociation of β-subunit from the α-subunit. Furthermore, alternative splicing can regulate BKCa activity resulting in varying calcium and hormone sensitivity of the channel.

BKCa is inhibited by the scorpion toxins charybdotoxin (ChTx) and by the highly selective iberiotoxin (ITX) (Figure 2). Both scorpion toxins block the pore occluding the K^+^ conduction of the α-subunit. In addition, there is a third scorpion toxin known to inhibit BKCa, called kaliotoxin. The tremorgenic mycotoxin paxilline is another high-affinity blocker of BKCa with a potency in the nano molar range. In addition, given extracellularly, BKCa is also very sensitive to tetraethylammonium with KD ~250 μM. Furthermore, several BK channel openers have been identified, including the synthetic benzimidazolone derivative NS1619, the compound NS11021, and the natural modulator dihydrosoyasaponin (Figure 3).

Understanding the physical structure of the BKCa is the key to gain insight into how the channels work. Therefore, the next chapters give an overview of the different subunits and their role in pulmonary circulation. 

## 4. The Role of the α-Subunit in the Pulmonary Circulation

The unique molecular protein design of the α-subunit of BKCa was described three decades ago [48,49]. In humans, the α-subunit is coded by a single gene-KCNMA1. Since its first description, there have been numerous publications showing the relevance in diseases. Favourable effects have been attributed to different activators of the channel. Cyclic AMP and protein kinase A are now well-established activators of BKCa by phosphorylating the channel protein in the pulmonary circulation [50]. The cytoplasmic tyrosine kinase c-Src inhibition also affect the BKCa channels open probability in hPASMC [51]. The synthetic benzimidazolone NS1619 has been shown to hyperpolarise pulmonary micro and macrovascular endothelial cells leading to cGMP-mediated endothelium-dependent vasodilation [52]. Inhaled NS1619 reduced the right ventricle pressure in an in vivo model of pulmonary hypertension and inhibited PASMC proliferation in vitro [53]. Furthermore, ex vivo investigations in the chronic hypoxia-induced PH rat model reported a significant vasodilation by NS1619 treatment. However, the precise binding site of NS1619 has not yet been determined. Furthermore, its relatively low potency and many off-target effects such as additional calcium channel inhibition limits its use. Instead, the synthetic biarylthiourea NS11021 has recently been established as a more selective and potent tool compound to study BK channel function. Like NS1619, NS11021 works by shifting the voltage-activation curve of the channel to more negative potentials reflected in increased open channel probability at single channel level [54].

As powerful vasodilators, prostacyclin analogues are important players in the treatment of severe pulmonary arterial hypertension (PAH) [55]. This pathway is mediated via the prostanoid I receptor-protein kinase A (PKA)-induced phosphorylation of BKCa in PASMCs and critically dependent on PPARβ/δ as a rapid signaling factor [47]. A naturally occurring polyunsaturated fatty acid docosahexanoic acid (DHA) has also been identified as a strong activator of BKCa, leading to a significant decrease of pulmonary arterial pressure [56,57]. For a long time, only the α-subunit had been taken into consideration for the DHA effect on the channel in endothelial cells. Detailed tissue-specific investigations unveiled that the DHA action is highly dependent on the co-expression of β-subunits and act via destabilizing the closed conformation of the ion conduction gate of the Slo1 + β1 channel [58]. Surprisingly, this activation does not require voltage-sensor activation or calcium-binding. Another natural substance of herbal origin, echinacoside, has been reported to activate BKCa in PAECs and PASMCs via the NO-cGMP-PKG axis [59]. Under chronic hypoxia, the steroid dehydroepiandrosterone (DHEA) prevented and reversed chronic pulmonary hypertension via activation of BKCa [60]. The effects of DHEA on BKCa is not cAMP or cGMP-dependent [61]. After cell injury, ischemia, or hypoxia, elevated levels of free heme due to their dissociation from the heme-bound protein complexes can be detected. Free heme has been shown to bind to the α-subunit of the BKCa channel, thereby inhibiting it [62]. In addition, protein kinase C and its isoenzymes, which reduce the levels of cAMP are known to deactivate BKCa [38]. Table 1 summarizes the studies carried out on modulation of the BKCa α-subunit in the pulmonary circulation.

Changes in the expression of several K^+^ channels have been described in pulmonary hypertension including the BKCa channel expression. However, the results are partly contradictory. In the hypoxia-induced PH rat model, downregulation of BKCa was detected [60]. Similarly, in the ovine model of the persistent pulmonary hypertension of the newborn, BKCa channel expression and function were reduced [66]. In contrast, BKCa was upregulated in the MCT-induced rat PH model [67]. Studies employing explanted lung samples from PH patients revealed more consistent results. Our study investigating laser-micro dissected pulmonary arteries obtained from IPAH patients showed upregulation of KCNMA1 in IPAH. Recently, a transcriptomic analysis has confirmed that BKCa is strongly upregulated in different forms of PAH [68]. The increased presence of BKCa channels in IPAH points to the possible utility of BKCa openers for pharmacological interventions in PAH. Accordingly, a recent preclinical study reported attenuation of the disease progression in the MCT-induced rat PH model by a selective BKCa opener [64]. In this study, also the elevated inflammatory TNF-α level was reduced, suggesting a dual effect of the drug in this PH model.

## 5. β-Subunit or the Modulatory Subunit of the BKCa Channel

Although the β-subunits are not uniformly expressed in every tissue in the body, their expression is precisely regulated [69]. Of the four β-subunits of BKCa channel (β1–β4) β1 is the most predominantly expressed β-subunit in the lung. Studies showing the presence and function of other β-subunits are scarce. β-subunits contain two transmembrane domains, intracellular N and C terminal ends and an extracellular loop responsible for interaction with specific inhibitors such as charybdotoxin. The coupling of the β1-subunit to the α-subunit, usually in a 1:1 ratio, increases the calcium sensitivity of the channel. In addition, this association is also known to increase the voltage sensitivity of the α-subunit [35,70,71,72,73]. Endogenous chemicals such as unsaturated free fatty acids, e.g., omega-3 docosahexaenoic acid, the sex hormone 17 β-estradiol or the human β-defensin 2 have been found to increase BKCa channel activity in an β1-subunit dependent manner, indicating a broad range of options to selectively manipulate BKCa [34].

In the lung, the β1 subunit expression is found mainly in the pulmonary artery smooth muscle cells, where it augments the channel′s sensitivity to Ca^2+^ sparks, contributing to pulmonary vasodilation [74]. Several mutations have been detected in the region of the gene encoding the β-subunit of the channel (KCNMB1) in humans. Fernandez-Fernandéz et al. found a mutation that resulted in a gain of function of the channel and showed its increased Ca^2+^-sensitivity. This mutation was associated with a low prevalence of severe diastolic hypertension [75]. In contrast, other mutations resulted in a loss-of-function of the channel. A polymorphism found in male African-Americans led to decreased open channel probability and was associated with a clinically significant decline of forced-expiratory volume (FEV) [76]. Animal studies knocking down the gene responsible for β-subunit show that these animals develop systemic hypertension and pulmonary hypertension under hypoxia [77]. The phenotype could be explained by the fact that hypoxia triggers the increase of the β1-subunit expression [78] through a mechanism involving HIF1-α [79]. In PASMCs of IPAH patients, several upregulated specific miRNAs against the β1-subunit have been identified. Overexpression of miR-29b in normal PASMC significantly downregulated KCNMB1 (BKCa β1-subunit) and decreased the channel activity. In contrast, inhibition of the same miRNA in IPAH-PASMCs enhanced the β1-subunit expression, the BKCa channel activity and thus, restored the phenotype [80]. Cao et al. detected decreased β1-subunit expression in the lung tissue of a small COPD cohort indicating that the β1-subunit might be related to the development of COPD. This study reported that miRNA-183 can decrease the β1-subunit protein expression in vascular smooth muscle cells [81]. This pathway could represent one possible mechanism of the pulmonary vascular phenotype in COPD [82]. Finally, in idiopathic pulmonary fibrosis (IPF), increased KCNMB1 expression correlated with the differentiation of fibroblasts into myofibroblasts, leading to exacerbation of the disease [83]. These studies support the relevance of the β1-subunit for regulating channel activity in the pulmonary vasculature pulmonary arteries and in the lung parenchyma and could resolve the molecular background of different clinical phenotypes. Table 2 provides more details of the research hitherto on the β1 subunit of the BKCa in the lung.

## 6. γ-Subunit or the Auxiliary Subunit of the BKCa Channel

The auxiliary γ1–4 subunits modulate the BKCa channel by shifting its voltage-dependence of the channel activation towards hyperpolarization. In the systemic circulation, leucine-rich repeat containing proteins (LRRC); a novel family of BK channel auxiliary γ-subunits, which are single membrane-spanning proteins containing a classic N-terminal cleavable signal peptide, have been identified recently [84]. However, in the lung, investigations on γ-subunit are predominantly concentrated on parenchymal cells. Recent investigation reported that the BKCa channel expression and function can be affected by inflammatory modulators. This has been substantiated by studies involving INF-γ treatment of the airway epithelial cells, which caused a downregulation of the γ-subunit, which in addition to the alteration of the expression of the other modulatory subunits reduced the channel activity [85]. In addition, downregulation of LRRC26 (γ1) and subsequent reduction of BKCa function have also been shown in cystic fibrosis bronchial epithelial cells. These findings correlated with the downregulation of the LRRC26 gene in the nasal cells in cystic fibrosis patients [86]. In cystic fibrosis patients, TGF-β1 is elevated and interferes with the expression of LRRC26. Mallotoxin, the BKCa opener in absence of LRRC26 was able to restore the ASL volume, indicating a potential for a novel therapeutic option [87]. Losartan, an angiotensin II receptor antagonist, also restored BKCa channel function by partial upregulation in the expression of the BKCa γ-subunit by alleviating the effect of TGF-β1 [88]. These studies clearly indicate the importance of the BKCa γ-subunit in the lung. However, our knowledge on this auxiliary subunits in the pulmonary circulation remains limited.

## 7. Perspective

BKCa channels are an important negative feedback system linking increases in intracellular calcium to outward potassium currents and membrane hyperpolarization. The experimental data suggest that BKCa channels play a crucial role in many pathophysiological conditions. Accordingly, multiple clinical trials on BKCa openers for cardiovascular diseases including hypertension, ischaemic heart disease, stroke, and erectile dysfunction have been initiated in the past decades. However, poor potency and the lack of selectivity of these compounds resulted in early discontinuation of these trials [89]. Only one drug candidate targeting BKCa channels (andolast) remained in the clinical development against mild asthma [90].

As described in the previous sections, BKCa could represent a promising molecular target for pulmonary hypertension therapy. There are cardiovascular drugs that involve the activation of BKCa channels, for instance NO or NO donors [64,91], PDE inhibitors such as sildenafil [92,93] or PGI2 analogues [47,94,95,96]. However, as BKCa is ubiquitously expressed throughout the vessels of the entire body, adverse side effects due to systemic vasodilation limit the use of systemically applied BKCa activators. Therefore, novel pathways, addressing specific properties of the target tissue, need to be developed. It shall then be possible to determine whether or not putative constituents identified within these pathways are druggable. The lung offers a special targeted approach by means of inhalation, which may provide pulmonary selectivity and even intrapulmonary selectivity to the best ventilated areas.

## Figures and Tables

**Figure 1 biomolecules-11-01629-f001:**
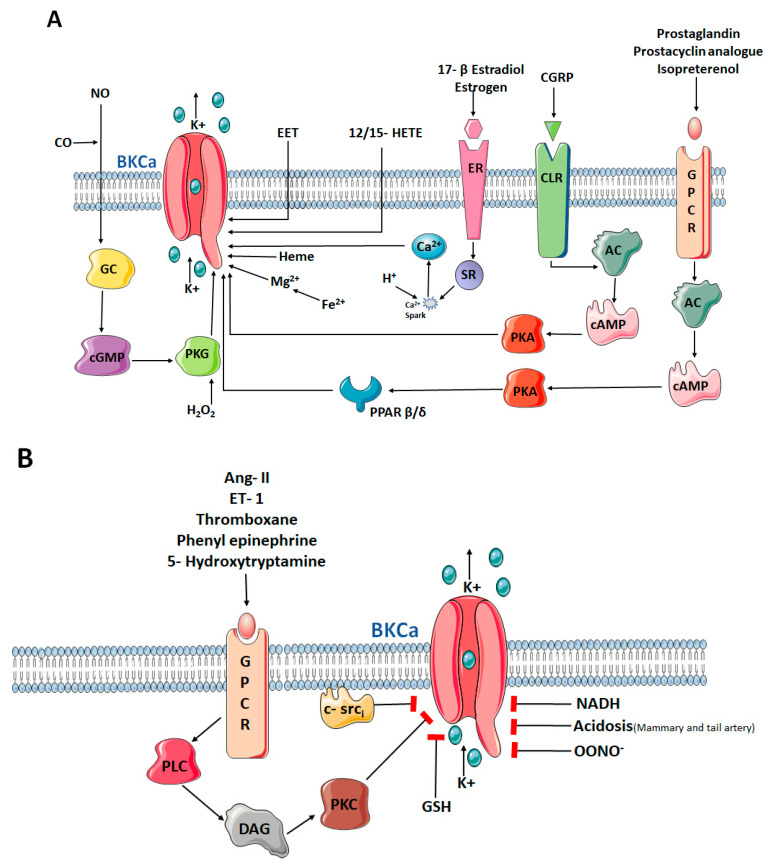
Intracellular and extracellular endogenous activators (**A**) and inhibitors (**B**) of BKCa channels. BKCa: Big/large conductance calcium activated potassium channels; NO: Nitric oxide; CO: Cabon monoxide; GC: Guanylyl cyclase; cGMP: Cyclic guanosine monophosphate; PKG: Protein Kinase G; H_2_O_2_: Hydrogen peroxide; K^+^: Potassium ion; Fe^2+^: Ferrous ion; Mg^2+^: Magnesium ion; EET: Epoxyeicosatrienoic acid; HETE: Hydroxyeicosatrienoic acid; ER: Estrogen receptor; SR: Sarcoplasmic reticulum; Ca^2+^: Calcium ion; GPCR: G-Protein coupled receptor; CGRP: Calcitonin gene related peptide; CLR: Calcitonin receptor-like receptor; AC: Adenylyl cyclase; PPAR: Peroxisome proliferator activated recetor; cAMP: Cyclic adenosine monophosphate; PKA: Protein kinase A. Ang II: Angiotensin II; ET-1: Endothelin-I; PLC: Phospholipase C; DAG: Diacylglycerol; PKC: Protein kinase C; c-srci: Tyrosine kinase src (inactivated); GSH: Glutatione; NADH: Reduced nicotinamide adenine dinucleotide; OONO^−^: Peroxynitrate. Figure created using the illustrations by Servier Medical Art by Srvier licensed under a Creative Commons Attribution 3.0 unported License.

**Figure 2 biomolecules-11-01629-f002:**
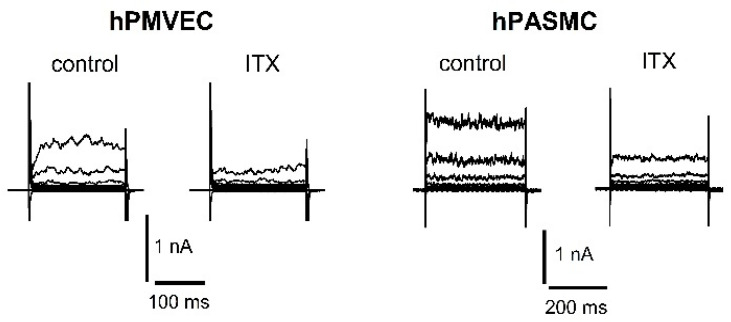
Representative patch-clamp recordings of K^+^ currents in primary human pulmonary microvascular endothelial cell (hPMVEC) and primary human pulmonary arterial smooth muscle cell (hPASMC). Recordings show control whole-cell K^+^ current and the reduction of the current after application of 100 nM ITX. In hPMVEC, currents were evoked from a holding potential of −50 mV using 200-ms voltage steps from −70 up to +90 mV in +20 mV increments. In hPASMC, currents were evoked from a holding potential of −60 mV using 400-ms voltage steps from −80 up to +80 mV in +20 mV increments. Experimental solutions have been described previously [47].

**Figure 3 biomolecules-11-01629-f003:**
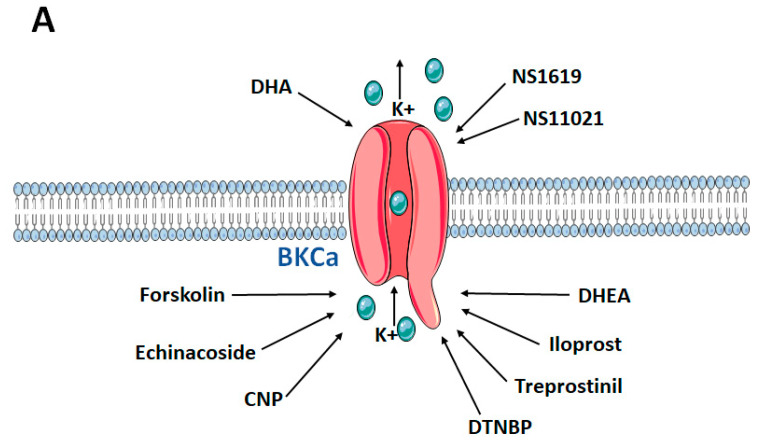
Exogenous activators (**A**) and inhibitors (**B**) of BKCa channels. BKCa: Big/large conductance calcium activated potassium channels; K+: Potassium ion; CNP: C-Type natriuretic peptide; DHA: Docosahexaenoic acid; NS1619: synthetic BKCa activator, DHEA: Dehydroepiandrosterone; DTNBP: Dimethyl dithiobispropionimidate; DTT: Dithiothreitol; TEA: Tetraethylammonium. Figure created using the illustrations by Servier Medical Art by Servier, licensed under a Creative Commons Attribution 3.0 unported License.

**Table 1 biomolecules-11-01629-t001:** Effect of pharmacological interventions of BKCa α-subunit in various disease models.

Compound	(Disease) Model	Effect on BKCa	Effects in the Model	Potential Mechanisms	References
NS1619 and C-type natriuretic peptide	Normotensive rats	Activation	Enhanced endothelium-dependent PA ring dilation and PA pressure reduction ex vivo; hyperpolarised and increased NO production in PMVECs in vivo	Direct activation of the channel in PMVECs	[63]
Compound X	Monocrotaline (MCT)-induced pulmonary arterial hypertension (PAH) rat model	Activation	Reduced pulmonary vascular remodelling, pulmonary flow resistance, RV hypertrophy and afterload in PAH model in vivo strongly vasodilated PA rings ex vivo	Direct activation of the channel in PASMCs	[64]
Docosahexaenoic acid (DHA)	IPAH patients normotensive rats chronic hypoxia-induced PH mouse model Kcnma1^−/−^ mouse	Activation	Reduced RV pressure in the PH animal model to normal in vivo enhanced endothelium-dependent PA ring dilation and PA pressure reduction ex vivo; dose-dependently activated BKCa current and hyperpolarised human IPAH-PASMCs to normal in vivo	Direct activation of the channel in PASMCs	[57]
Dehydroepiandrosterone (DHEA)	Chronic hypoxia-induced PH rat model	Activation	Reduced RV pressure, RV wall thickness and PA remodelling in the PH animal model in vivo restored the PA-pressure response to acute hypoxia in the PH animal model in vivo. Decreased intracellular [Ca^2+^] under hypoxia in PASMCs in vitro Increased BKCa channel activity and expression in PA in the PH animal model in vitro	Dual effect: (i) Activation of the channel in PA by changing the redox balance toward a more oxidative state (ii) Upregulated BKCa mRNA and protein in PASMCs of the chronic hypoxic PH model	[60]
Echinacoside	Normotensive rats	Activation	Reduced noradrenaline-induced contraction of PA in extracellular [Ca^2+^]-dependent manner ex vivo	Activated NO-cGMP-PKG pathway with subsequent hyperpolarisation and decrease of intracellular free [Ca^2+^] in PASMCs	[59]
Forskolin and cAMP activators	Fawn-hooded rat employed in chronic hypoxia-induced PH rat model	Activation	Increased open probability of BKCa channels in fawn-hooded PH animal PASMCs in vitro	Direct activation of the channel in PASMCs	[50]
NS1619	Monocrotaline (MCT)-induced PAH rat model	Activation	Reduced RV pressure, carbon monoxide and improved oxygenation in the PAH animal model in vivo reduced PDGF-induced PASMC proliferation in vitro	Direct activation of the channel	[53]
JAK2 inhibitors	IPF patients bleomycin-induced lung fibrosis and PH rat model	Activation	Reduced RV pressure and PA Remodelling in vivo Reduced V/Q mismatch in animal model in vivo Promoted relaxant and anticontractile effects on IPF-PA ex vivo Inhibited effect of TGFβ1-induced loss of endothelial markers and upregulation of the PA remodelling markers in vitro Activated BKCa current in vitro	Unknown	[65]
Iloprost and treprostinil	Primary human PASMCs	Activation	Enhanced PA ring dilation ex vivo Enhanced BKCa current in vitro	(PKA)-induced phosphorylation of BKCa	[47]
NS1619	Chronic hypoxia-induced PH rat model	Activation	Enhanced NO-dependent PA pressure reduction ex vivo	Direct activation of the channel	[52]

**Table 2 biomolecules-11-01629-t002:** Effect of interventions of BKCa β1-sub unit in various disease models.

Compound	(Disease) Model	Effect on BKCa	Effects in the Model	Potential Mechanisms	References
Knocking down KCNMB1	Chronic hypoxia-induced PH mouse model employing Kcnmb1^−/−^ mouse	Inhibition	Increased pulmonary vascular response to acute and chronic hypoxia and increased RV pressure in vivo Increased focal complex expression in PASMCs in vitro	Downregulates BKCa channels (mRNA, protein and function) in PASMCs	[77]
Knocking down HIF-1α or KCNMB1	Subacute hypoxia in hPASMCs	Inhibits upregulation of KCNMB1 in response to hypoxia	Potentiated the hypoxic-mediated increase in [Ca^2+^]_i_ in vitro Strongly vasodilated PA rings ex vivo	Downregulates BKCa channels (mRNA, protein and function) in PASMCs	[79]
Overexpression or inhibition of miR-29b	Healthy and IPAH PASMCs	Inhibition or activation, respectively	Decreased BKCa channel currents and downregulated KCNMB1 in normal PASMCs in vitro Increased BKCa channel activity and upregulated KCNMB1 in IPAH-PASMC in vitro	Downregulation or activation of the channel in PASMCs	[80]
Upregulated KCNMB1	IPF fibroblasts	Activation	Increased BKCa channel activity in vitro	Upregulated BKCa mRNA and protein in IPF fibroblasts	[83]

## Data Availability

Not applicable.

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
