# Peer review of "Revisiting the Large-Conductance Calcium-Activated Potassium (BKCa) Channels in the Pulmonary Circulation"

_biomolecules, 2021, doi:10.3390/biom11111629_

Round 1
Reviewer 1 Report
In their review article Guntur and colleagues present their view on large conductance potassium (BKCa) channels in the pulmonary vasculature and their potential role under pathophysiological conditions. Authors have a long track expertise on the role the role of K+ channels in the pulmonary circulation. With the present paper, they aim to revisiting a family of K channels that has been quite disregarded in the last years despite their potential as therapeutic target in pulmonary vascular diseases. In my opinion this is a very well organised, comprehensive, and updated review.
I have several comments and suggestions:
1. In Figure 1 the protocol applied to record the K+ currents should be included in the figure legend or illustrated in the figure. This is important to show the range of test potentials where the macroscopic current is observed.
2. Throughout the review, some illustrative cartoons are missing. For instance, a cartoon showing the main vasoactive factors targeting BKCa channels present in both PASMC and PAECS from the pulmonary circulation (i.e. PGI2, NO, ANP, etc…) would be very helpful for the readers as a summary of the detailed information included the manuscript.
3. As the authors know, downregulation of several K+ channels (especially, KCNK3 and KCNA5) is a common feature in pulmonary hypertension. This does not seem to be the case for BKCa channels. Thus, authors have previously shown that BKCa is upregulated in idiopathic pulmonary arterial hypertension (PMID: 27540020). Recently a transcriptomic analysis (PMID: 34349187) has also revealed that BKCa is strongly upregulated in different forms of PAH, confirming the above data. I recommend including more information on the expression of BKCa channel subunits in the context con experimental and clinical PAH and how this may influence the effects of BKCa agonists.
4. Since pulmonary arterial hypertension has a well-established inflammatory component authors could mention or discuss the potential anti-inflammatory effects induced by BKCa channel opening. There are some studies pointing on that direction (PMID: 33245463; PMID: 23995289).
Author Response
We would like to thank the reviewer for the valuable time in reviewing our manuscript.
As requested, we have made textual and graphical adaptations. We believe these edits have improved our manuscript significantly. Please find attached the updated manuscript, wherein the changes are marked using the track-changes function of Microsoft Word.
In their review article Guntur and colleagues present their view on large conductance potassium (BKCa) channels in the pulmonary vasculature and their potential role under pathophysiological conditions. Authors have a long track expertise on the role the role of K+ channels in the pulmonary circulation. With the present paper, they aim to revisiting a family of K channels that has been quite disregarded in the last years despite their potential as therapeutic target in pulmonary vascular diseases. In my opinion this is a very well organised, comprehensive, and updated review.
I have several comments and suggestions:
1. In Figure 1 the protocol applied to record the K+ currents should be included in the figure legend or illustrated in the figure. This is important to show the range of test potentials where the macroscopic current is observed.
Thank you for this comment! The figure legend has been extended by the protocol.
2. Throughout the review, some illustrative cartoons are missing. For instance, a cartoon showing the main vasoactive factors targeting BKCa channels present in both PASMC and PAECS from the pulmonary circulation (i.e. PGI2, NO, ANP, etc…) would be very helpful for the readers as a summary of the detailed information included the manuscript.
We have extended the manuscript by illustrative cartoons.
3. As the authors know, downregulation of several K+ channels (especially, KCNK3 and KCNA5) is a common feature in pulmonary hypertension. This does not seem to be the case for BKCa channels. Thus, authors have previously shown that BKCa is upregulated in idiopathic pulmonary arterial hypertension (PMID: 27540020). Recently a transcriptomic analysis (PMID: 34349187) has also revealed that BKCa is strongly upregulated in different forms of PAH, confirming the above data. I recommend including more information on the expression of BKCa channel subunits in the context con experimental and clinical PAH and how this may influence the effects of BKCa agonists.
The revised manuscript discusses the expression of the BKCa channel in the context of experimental and clinical PAH and its potential relevance for PAH therapy.
4. Since pulmonary arterial hypertension has a well-established inflammatory component authors could mention or discuss the potential anti-inflammatory effects induced by BKCa channel opening. There are some studies pointing on that direction (PMID: 33245463; PMID: 23995289).
Thank you for this comment! The revised manuscript discusses the anti-inflammatory potential of BKCa openers in PH.

Reviewer 2 Report
The review is interesting and comprehensive
Minor remarks
- One or two figures (as graphical abstract) may improve the quality of this review
- The authors could add one sentence on the localization of BKCa, and more precisely on a putative presence on mitochondria and so putative new functions.
- Author could add a short paragraph on the interaction of BKCa channels with others calcium channels like ryanodine (in complement to the line 160).
- Line 56: autonomous nervous system does not control PVR. This affirmation is conflictual and the sentence should be re-written. See publication PMID: 29202826
- Line 126 : add perhaps the synonym Slo1
Author Response
We would like to thank the reviewer for the valuable time in reviewing our manuscript.
As requested, we have made textual and graphical adaptations. We believe these edits have improved our manuscript significantly. Please find attached the updated manuscript, wherein the changes are marked using the track-changes function of Microsoft Word.
One or two figures (as graphical abstract) may improve the quality of this review
As also requested by reviewer 1, we added a few figures.
The authors could add one sentence on the localization of BKCa, and more precisely on a putative presence on mitochondria and so putative new functions.
We have added a new paragraph extensively discussing intracellular BKCa.
Author could add a short section on the interaction of BKCa channels with others calcium channels like ryanodine (in complement to the line 160).
We have extended the manuscript accordingly.
Line 56: autonomous nervous system does not control PVR. This affirmation is conflictual and the sentence should be re-written. See publication PMID: 29202826
Thank you for this comment. It is not an easy topic. We have revised and extended this paragraph.
Line 126 : add perhaps the synonym Slo1
Slo1 is now added.
